# COX-2/sEH Dual Inhibitor Alleviates Hepatocyte Senescence in NAFLD Mice by Restoring Autophagy through Sirt1/PI3K/AKT/mTOR

**DOI:** 10.3390/ijms23158267

**Published:** 2022-07-27

**Authors:** Chen-Yu Zhang, Xiao-Hua Tan, Hui-Hui Yang, Ling Jin, Jie-Ru Hong, Yong Zhou, Xiao-Ting Huang

**Affiliations:** 1Department of Physiology, School of Basic Medicine Science, Central South University, Changsha 410078, China; zhangchenyu@csu.edu.cn (C.-Y.Z.); enzezeen@163.com (H.-H.Y.); xyjinling@126.com (L.J.); hongjieru2020@163.com (J.-R.H.); zhouyong421@csu.edu.cn (Y.Z.); 2Experimental Center of Medical Morphology, School of Basic Medicine Science, Central South University, Changsha 410078, China; 213131@csu.edu.cn; 3Xiangya Nursing School, Central South University, Changsha 410078, China

**Keywords:** non-alcoholic fatty liver disease, COX-2/sEH dual inhibitor, senescence, autophagy

## Abstract

We previously found that the disorder of soluble epoxide hydrolase (sEH)/cyclooxygenase-2 (COX-2)-mediated arachidonic acid (ARA) metabolism contributes to the pathogenesis of the non-alcoholic fatty liver disease (NAFLD) in mice. However, the exact mechanism has not been elucidated. Accumulating evidence points to the essential role of cellular senescence in NAFLD. Herein, we investigated whether restoring the balance of sEH/COX-2-mediated ARA metabolism attenuated NAFLD via hepatocyte senescence. A promised dual inhibitor of sEH and COX-2, PTUPB, was used in our study to restore the balance of sEH/COX-2-mediated ARA metabolism. *In vivo*, NAFLD was induced by a high-fat diet (HFD) using C57BL/6J mice. *In vitro*, mouse hepatocytes (AML12) and mouse hepatic astrocytes (JS1) were used to investigate the effects of PTUPB on palmitic acid (PA)-induced hepatocyte senescence and its mechanism. PTUPB alleviated liver injury, decreased collagen and lipid accumulation, restored glucose tolerance, and reduced hepatic triglyceride levels in HFD-induced NAFLD mice. Importantly, PTUPB significantly reduced the expression of liver senescence-related molecules p16, p53, and p21 in HFD mice. *In vitro*, the protein levels of γH2AX, p53, p21, COX-2, and sEH were increased in AML12 hepatocytes treated with PA, while Ki67 and PCNA were significantly decreased. PTUPB decreased the lipid content, the number of β-gal positive cells, and the expression of p53, p21, and γH2AX proteins in AML12 cells. Meanwhile, PTUPB reduced the activation of hepatic astrocytes JS1 by slowing the senescence of AML12 cells in a co-culture system. It was further observed that PTUPB enhanced the ratio of autophagy-related protein LC3II/I in AML12 cells, up-regulated the expression of Fundc1 protein, reduced p62 protein, and suppressed hepatocyte senescence. In addition, PTUPB enhanced hepatocyte autophagy by inhibiting the PI3K/AKT/mTOR pathway through Sirt1, contributing to the suppression of senescence. PTUPB inhibits the PI3K/AKT/mTOR pathway through Sirt1, improves autophagy, slows down the senescence of hepatocytes, and alleviates NAFLD.

## 1. Introduction

Non-alcoholic fatty liver disease (NAFLD) is a liver disease characterized by hepatic fat accumulation, inflammation, and hepatic cell dysfunction, which is closely related to obesity and metabolic syndrome [1]. NAFLD could progress to fibrosis, cirrhosis, and end-stage liver failure [2]. The clinical case of advanced cirrhosis is mainly irreversible [3]. Although there are many studies on NAFLD, the incidence of NAFLD is still as high as 25–30%, due to the complex pathogenesis and lack of effective treatment [4]. Therefore, new and effective strategies are needed to delay or reverse NAFLD.

Arachidonic acid (ARA) is one of the most abundant lipid molecules in the body. Its metabolites have a variety of biological functions and are widely involved in the physiological and pathological processes of the body [5]. Intracellular free ARA can generate epoxyeicosatrienoic acid (EETs) under the action of cytochrome P450 (CYP) epoxygenases [6]. EETs have a variety of biological activities, such as anti-inflammatory [7,8], anti-oxidant [9], and anti-fibrosis [10], while EETs are quickly inactivated by soluble epoxide hydrolase (sEH) [11]. ARA can also be transformed into prostaglandin under the action of cyclooxygenase-2 (COX-2) [12]. We have previously reported that the disorder of sEH and COX-2-mediated ARA metabolism involves acute lung injury, pulmonary fibrosis, and sepsis [13,14,15]. Recently, we found a significant increase in the expression of sEH and COX-2 proteins in the liver of NAFLD mice, manifesting as abnormal ARA metabolism [16], while a dual inhibitor of sEH/COX-2, PTUPB, could reduce lipid droplet accumulation and inflammatory response in the liver and hepatocytes of NAFLD mice [16]. However, the exact mechanism underlying PTUPB alleviating NAFLD needs further study.

Increasing evidence points out that cellular senescence accelerates the progression of several diseases, such as atherosclerosis, neuropathy, and NAFLD [17,18,19]. Cellular senescence is an irreversible and cycle-retarded evolutionary conserved state [11]. Senescent cells can synthesize and secrete a large number of senescence-related secretory phenotypes (SASP), accompanied by the down-regulation of intracellular anti-senescence molecules, such as Sirt1 [20,21]. It has been found that the liver cells of NAFLD patients showed senescence, with severe DNA damage and cell cycle arrest [18]. Moreover, the aggravation of steatosis in high-fat diet (HFD) rats was accompanied by increased expression levels of p16 and p21 in liver tissue [22]. The accumulation of senescent cells has been shown to promote lipid accumulation and steatosis of hepatocytes [18]. Moreover, senescent hepatocytes could enhance the activation and expression of pro-fibrotic molecules in co-cultured primary hepatic stellate cells [23]. Eliminating the p16^+^ senescent cells by gene editing or anti-senescent drugs (dasatinib and quercetin) could reduce the number of senescent cells and attenuate liver lipid deposition in db/db mice, suggesting that the number of senescent cells is positively correlated with liver lipid deposition [18]. Therefore, slowing cellular senescence may be a promising strategy for the treatment of NAFLD.

Studies have shown that EETs could reduce senior-induced dysfunction of arterial endothelial cells [24]. Overexpression of CYP2J2 could increase the generation of EETs and reduce the protein expressions of p53 and p16, markers of endothelial cell senescence induced by hydrogen peroxide, and reduce the number of β-galactosidases (β-gal) staining positive senescent cells [25]. Inhibition of sEH activity significantly reduced the mRNA and protein expression of p53 and p21 in the lung of pulmonary fibrosis mice [14]. However, the generation of endogenous EETs decreases in the aging body [26]. These results suggest that EETs may be an essential endogenous anti-senescence molecule. In addition, the expression of COX-2 is increased in senescent astrocytes [27]. Overexpression of COX-2 is associated with neurodegenerative diseases in the elderly [28]. These studies suggest that sEH/COX-2-mediated ARA metabolic disorder is closely related to cellular senescence. However, it is not clear whether PTUPB can reduce hepatocyte senescence in NAFLD.

Autophagy refers to transporting damaged, aging, or denatured proteins and organelles to the lysosomes for degradation [29]. It is an adaptive reaction of cells under the action of various adverse factors to achieve the purpose of recycling materials and energy, and maintaining the stability of the internal environment of cells [30]. Less efficient autophagy with age leads to a build-up of harmful proteins and damages cells [31]. Therefore, aging is closely related to the reduction in autophagy activity. Additionally, autophagy also has an anti-aging effect, which can prolong the life span of many model organisms [32]. It has been reported that the sEH knockout restores HFD-induced lipotoxic cardiomyopathy and mTOR signaling-mediated cardiac autophagy dysfunction [33]. Exogenous 11,12-EET significantly restores the ethanol-induced cardiac autophagy flux injury in neonatal rats [34]. Meanwhile, it has been found that inhibition of COX-2 in aging-related Parkinson’s disease could up-regulate the autophagy of nerve cells [35]. Therefore, restoring autophagy levels by regulating ARA metabolism is meaningful in treating age-related diseases.

In this study, we investigate whether PTUPB can reduce obesity-induced hepatocyte senescence, and explore the potential mechanism of PTUPB’s anti-senescence by restoring autophagy.

## 2. Results

### 2.1. PTUPB Attenuates HFD-Induced NAFLD in Mice

Firstly, we explored the inhibitory effect of COX-2/sEH dual inhibitor PTUPB (5 mg/kg, *s.c.* once a day) on HFD-induced NAFLD in mice (Figure 1A). Compared with the Control group, the mice in the HFD group showed noticeable histopathological changes in liver tissue, including the damage to the liver structure and abundant inflammatory cell infiltration, which was reduced by PTUPB treatment (Figure 1B). Oil Red O staining results showed that PTUPB treatment significantly reduced the red lipid droplets in liver tissue of HFD mice (Figure 1B). Masson staining results showed that HFD feeding for 12 weeks resulted in a large amount of collagen deposition in the liver of mice, mainly in the liver lobule and around the central lobule vein. PTUPB treatment significantly reduced the collagen content in mice livers (Figure 1B), which was confirmed by the Western blot results of collagen III (Figure 1C,D). These were consistent with our previously published findings [16]. Then we took it a step further. Compared with the control group, the glucose tolerance of HFD mice was significantly reduced, while PTUPB significantly increased the glucose tolerance of HFD mice. Further analysis of the area under the curve (AUC) showed the same results (Figure 1E,F). The liver triglyceride (TG) level of mice in the HFD group was significantly higher than that in the Control group. PTUPB treatment significantly reduced the content of TG in the liver of NAFLD mice (Figure 1G). Collectively, these results suggest that PTUPB attenuates HFD-induced NAFLD in mice.

### 2.2. PTUPB Inhibits Hepatocyte Senescence Induced by HFD in Mice

We were surprised that the mRNA expression of senescence-related molecules *p16* and *p19* significantly increased in the liver of HFD mice, which was reduced by PTUPB treatment (Figure 2A,B). Furthermore, we found that the protein expression of p53 and p21 in the liver of HFD mice was also significantly enhanced, and PTUPB treatment decreased the expressions of p53 and p21 (Figure 2C–E). We also observed that PTUPB reduced the expression of p16 and p21 proteins in the liver of HFD mice by immunofluorescence (Figure 2F–H). These results suggest that PTUPB treatment inhibits the senescence of liver tissue in HFD-induced NAFLD mice.

### 2.3. PTUPB Attenuates PA-Induced Hepatocyte Senescence In Vitro

To illustrate the role of senescence in protecting PTUPB against NAFLD, we investigated the effects of PTUPB on hepatocyte senescence *in vitro*. We stimulated mouse hepatocyte cell line (AML12) with palmitic acid (PA) for 48 h and found that 200 μM PA significantly inhibited the proliferation of AML12 cells. Immunofluorescence results showed that the expression levels of Ki67 and PCNA were significantly decreased, while 2000 μM PA induced a large number of cell deaths, suggesting that 200 μM PA could induce cell cycle arrest (Figure 3A). We also found that 200 μM PA treatment significantly up-regulated the senescence-related protein expressions of p53, p21, and γH2AX in AML12 cells (Figure 3B,C–E). Importantly, PA treatment increased the protein expressions of COX-2 and sEH in AML12 cells (Figure 3B,F,G), indicating that dysregulation of ARA metabolism participates in the development of PA-induced hepatocyte senescence. Next, we found that PTUPB remarkably decreased PA-induced lipid deposition in AML12 cells (Figure 3H), reduced the intensity of positive senescence-associated β-galactosidase (SA-β-gal) staining (Figure 3I), and significantly inhibited the expression of p53, p16, and γH2AX (Figure 3J–M). Altogether, these results indicate that PTUPB pretreatment attenuates PA-induced hepatocyte senescence *in vitro*.

### 2.4. PTUPB Reduces the Hepatic Astrocytes Activation by Inhibiting Hepatocyte Senescence

In the process of liver fibrosis, hepatic astrocytes are essential effector cells, which can be transdifferentiated into myofibroblasts with proliferative, contractile, and secretory properties. We used mouse hepatic astrocytes (JS1) to evaluate the effect of PA-induced senescent hepatocytes on the activation of hepatic astrocytes. AML12 cells were stimulated with or without PA, and the cell culture supernatant was collected as a conditioned medium (CM). We found that 20% CM significantly increased the expression of collagen I, collagen III, and α-SMA proteins in JS1 cells (Figure 4A–D). While the CM from the AML12 cells treated by PA with PTUPB (PA+PTUPB-CM) failed to induce the collagen I and α-SMA proteins in JS1 cells (Figure 4E–G). Those findings indicate that PTUPB reduces hepatic astrocyte activation by inhibiting hepatocyte senescence.

### 2.5. PTUPB Attenuates Hepatocyte Senescence by Enhancing Autophagy

Then, we wondered how PTUPB attenuated PA-induced hepatocyte senescence. Impaired autophagy is one of the markers of senescence. We found that the expression of autophagy-related protein LC3 was significantly decreased in the liver of HFD mice, which was restored after PTUPB treatment (Figure 5A,B). *In vitro*, we found that the LC3II/LC3I ratio and the expression of mitochondrial autophagic protein Fundc1 were significantly decreased in PA-treated AML12 cells, while the autophagic substrate protein p62 was increased, indicating impaired autophagy. PTUPB treatment almost restored the expression ratio of LC3II/LC31 and Fundc1 in PA-treated AML12 cells and reduced p62 protein (Figure 5C–F). Further, to elucidate the role of autophagy in the inhibitory effects of PTUPB against senescence, we used an autophagy inhibitor, 3-MA. The results showed that the inhibition of autophagy by 3-MA reversed the anti-senescence effect of PTUPB (Figure 5G–L). These results suggest that PTUPB enhances autophagy in PA-treated hepatocytes, contributing to the anti-senescence effect.

### 2.6. PTUPB Enhances Hepatocyte Autophagy by Promoting Sirt1 Expression

Sirt1 is a crucial anti-senescence molecule. We found that Sirt1 protein expression was decreased in HFD-fed mouse liver and PA-stimulated AML12 cells, and PTUPB restored the Sirt1 expression (Figure 6A–D). To determine whether Sirt1 plays a vital role in the anti-senescence process of PTUPB, we used a Sirt1 inhibitor, EX527. We found that the anti-senescence effect of PTUPB was abolished by EX527, characterized by the increased expression of γH2AX (Figure 6E,G). Additionally, EX527 also abolished the enhanced autophagy by PTUPB pre-treatment, characterized by the decreased Fundc1 protein and LC3II/LC3I ratio (Figure 6E,H–I). These results suggest that PTUPB restores autophagy via Sirt1 and alleviates PA-induced senescence of AML12 cells.

### 2.7. PTUPB Inhibits the PI3K/AKT/mTOR Signaling Pathway by Promoting Sirt1 Expression

One of the core events of autophagy begins with the inhibition of mTOR. We found that total mTOR protein and phosphorylation levels were significantly increased in HFD-fed mice liver, while PTUPB inhibited the increase in mTOR protein and its phosphorylation level (Figure 7A–C). At the same time, in the senescent AML12 cells induced by PA, we found that the phosphorylation level of mTOR was enhanced, and PTUPB treatment reduced the phosphorylation level of mTOR (Figure 7D–G). Meanwhile, we found that the phosphorylation of PI3K and AKT, the upstream of mTOR, was enhanced, which was inhibited by PTUPB (Figure 7D–G), suggesting that PTUPB inhibits the activation of the PI3K/AKT/mTOR signaling pathway. To determine whether Sirt1 plays a role in PTUPB’s inhibition of the PI3K/AKT/mTOR signaling pathway, we further applied the Sirt1 inhibitor EX527. The results showed that PTUPB’s inhibition of the PI3K/AKT/mTOR signaling pathway was reversed by EX527 (Figure 7H–K). These results indicate that PTUPB inhibits the activation of the PI3K/AKT/mTOR signaling pathway through Sirt1.

## 3. Discussion

This study demonstrates, for the first time, that PTUPB attenuates HFD-induced NAFLD by inhibiting hepatocyte senescence *in vivo* and *in vitro*. Our study showed that cellular senescence occurred in the liver of HFD mice, and PTUPB reduced hepatocyte senescence in NAFLD mice by restoring the sEH/COX-2 metabolic homeostasis of ARA. Further *in vitro* studies have found that PTUPB’s anti-senescence effect is through promoting Sirt1 expression and restoring autophagy. Our study indicates that dysregulated metabolism of CYPs/COX-2-derived ARA plays an essential role in the process of hepatocyte senescence in NAFLD.

More and more studies indicate that cellular senescence is involved in the pathogenesis of NAFLD. Although senescent cells cannot proliferate, they are still metabolically active and communicate with surrounding cells, thereby spreading senescence and inducing damage through SASP [36]. Some studies have proposed that hepatocyte senescence may be a central pathological mechanism that can promote intracellular fat accumulation, fibrosis and inflammation, and secretory inflammatory mediators related to senescence [37]. Our study also found that the liver of NAFLD mice showed signs of senescence, and the AML12 cells stimulated by PA showed senescence, with high expression of p16, p21, and γH2AX proteins, accompanied by increased SA-β-Gal activity. To further clarify the mechanism by which sEH/COX-2 metabolism disorders accelerate NAFLD, we observed sEH/COX-2 metabolism *in vitro*, which was consistent with the tissue level of our previous study, showing significantly increased protein expressions of sEH and COX-2. PTUPB decreased the expression of senescence-related proteins in AML12 cells induced by PA. These results indicate that PTUPB could slow down the senescence of hepatocytes.

There is increasing evidence that autophagy plays a significant role in obesity and metabolic regulation [38]. It has been found that dysregulation of liver autophagy occurs after chronic HFD treatment [39]. The autophagic activity was inhibited due to a significant reduction in the expression of autophagy-related genes, such as in a mouse model and in patients with fatty liver [40]. Our results also indicated that autophagy level was reduced in NAFLD, manifested by the decreased LC3II/LC3I ratio, the accumulation of autophagy substrate p62 protein, and the decreased expression of mitochondrial autophagy-related protein Fundc1 in PA-treated AML12 cells, indicating decreased liver autophagy level in NAFLD. It has been reported that H_2_S could activate liver autophagy, reducing serum TG and alleviating NAFLD [41]. Further studies found that HFD-induced NAFLD could be improved by enhancing autophagy by inhibiting the ROS/PI3K/AKT/mTOR signaling pathway [42]. These results suggest that enhanced autophagy alleviates NAFLD. Senescence can lead to decreased autophagy activity, associated with impaired intracellular component accumulation and subsequent disruption of homeostasis and dysfunction [43]. Studies have shown that senescent AML12 cells show increased mTOR signaling and decreased autophagy levels [44]. Emerging evidence suggests that activation of autophagy can prevent age-related diseases and promote longevity. For example, inhibition of mTORC1/TFEB signaling restores autophagy activity and slows down liver aging [45]. These results suggest that inhibition of the mTOR signaling pathway can enhance autophagy and reduce liver aging and NAFLD. In our study, we found that PTUPB could significantly restore the autophagy level of senescent hepatocytes, while the autophagy inhibitor 3-MA could counteract the anti-senescence effect of PTUPB. Further, we found that PTUPB could inhibit the activation of the PI3K/AKT/mTOR signal upstream of autophagy. Thus, it is reasonable to speculate that targeted activation of autophagy is a pivotal contributor to the liver-protective activity of PTUPB.

Sirt1, as a critical intracellular anti-senescence molecule, has attracted much attention. Overexpression or activation of Sirt1 can inhibit liver aging [46]. Studies have shown that Sirt1 could activate autophagy through a variety of pathways, including regulation of fox headbox transcription factor (FOX), ATG5, ATG7, LC3, and Beclin1 [47,48]. FGF21-induced autophagy flux enhancement was mediated by the nuclear translocation of TFEB, which occurs due to the activation of the Sirt1-mTOR signaling pathway [49]. In this study, we found that PTUPB can stabilize Sirt1 protein in liver and hepatocytes. Sirt1 inhibitor EX527 partially eliminated PTUPB’s inhibitory effect on the PA-stimulated mTOR signaling pathway in hepatocytes, suggesting that PTUPB inhibits the PI3K/AKT/mTOR signaling pathway through Sirt1.

In conclusion, our findings suggest that the sEH/COX-2 metabolic disorder of ARA plays an essential role in the senescence process of hepatocytes in NAFLD mice. PTUPB inhibits the PI3K/AKT/mTOR signaling pathway via Sirt1, which restores autophagy and inhibits the hepatocytes senescence in NAFLD (Figure 8). This study might promote the application of PTUPB in NAFLD treatment.

## 4. Materials and Methods

### 4.1. Animal Experiments

Male C57BL/6 mice (8 to 10 weeks of age, 20–25 g) were provided by Hunan SJA Laboratory Animal Co. Ltd. (Hunan, China). The animal experiment was approved by the Ethics Committee of Hunan University of Medicine, 2021 (A0819006).

### 4.2. HFD Mouse Model

C57BL/6J mice were randomly divided into Control, PTUPB, HFD, and HFD+PTUPB groups, with 8 mice in each group. Mice in the HFD and HFD+PTUPB groups were given a 60% high-fat and high-sugar diet. Mice in the Control and PTUPB groups were given a regular diet. PTUPB group and HFD+PTUPB mice were subcutaneously injected with PTUPB (5 mg/kg) daily, while mice were administered PGE400 in the Control and HFD groups. PTUPB was given by Bruce D. Hammock at UC Davis Comprehensive Cancer Center, University of California [50]. Twelve weeks later, the mice were anesthetized with an intraperitoneal injection of pentobarbital sodium (80 mg/kg) for subsequent experiments [51].

### 4.3. Liver Histopathology Analysis

The liver tissue was conventionally paraffin-embedded or frozen sectioned. Paraffin-embedded sections were stained with HE and Masson. The pictures were detected by a microscope (Motic, BA410E, Motic China group CO. LTD., Hong Kong, China).

### 4.4. Triglycerides Measurement

Mouse liver tissue was ground in the ice bath to prepare homogenate. The content of TG in liver tissue was determined by a TG assay kit (Jiancheng Bioengineering Institute, Nanjing, China).

### 4.5. Cell Culture and Treatment

Mouse hepatocyte cell line (AML12) and mouse hepatic astrocyte line (JS1) were purchased from CAS Shanghai Cell Bank (Shanghai, China). AML12 cells were cultured in DMEM/F12 medium (containing 10% fetal bovine serum, Gibco, Grand Island, NY, USA), Insulin-Transferrin-Selenium supplement (Gibco), and 40 ng/mL dexamethasone (Sigma-Aldrich, Burligton, MA, USA). JS1 cells were cultured in DMEM/F12 medium (containing 10% fetal bovine serum) at 37 °C incubators containing 5% CO_2_.

AML12 cells were planted into the plate. Then cells were stimulated with palmitic acid (PA, 200 μM, Sigma-Aldrich, Burligton, MA, USA) for 48 h and collected for detection. AML12 cells were pre-treated with PTUPB for 1 h to observe the effect of PTUPB on PA-induced senescence of AML12 cells. An autophagy inhibitor 3-MA (5 mM, MCE, Dallas, TX, USA) or Sirt1 inhibitor EX527 (10 μM, Topscience, Shandong, China) was used to clarify the role of autophagy or Sirt1 in the anti-senescence effects of PTUPB in AML12 cells.

### 4.6. Oil Red O Stain

The frozen slices were rewarmed at room temperature for 10 min, then washed with distilled water. For cell samples, the cells were washed twice with PBS, fixed with 4% paraformaldehyde for 30 min, and washed twice with distilled water. The slices or cells were immersed in 60% isopropyl alcohol for 2 min. Then the Oil Red O working solution was dyed, and the color was toned with 60% isopropyl alcohol after 5 min, and washed immediately after toning. Hematoxylin was stained for 1 min, then rinsed under running water for 10 min for anti-blue. The tablets were sealed with glycerin gelatin, observed under an inverted microscope (Motic, BA410E, Motic China group CO. LTD., Hong Kong, China), and photographed.

### 4.7. Senescence-Associated β-Galactosidase Staining

SA-β-gal staining kit was obtained from Beyotime Biotechnology (Shanghai, China). The cells were washed with PBS three times, and 0.6 mL fixed buffer was added to each well for 15 min at room temperature. The cells were again washed with PBS three times. The staining mixture was added and then incubated overnight at 37 °C. The next day, cells were washed with PBS and observed under a microscope (Nikon, Tokyo, Japan).

### 4.8. Glucose Tolerance Measurement

Mice were fasted for 12 h and given an intraperitoneal injection of glucose (2 g/kg body weight). The blood samples were collected by tail vein to measure the blood glucose concentration at 0, 15, 30, 60, and 120 min after glucose injection.

### 4.9. Western Blot

The liver tissue or cells were added to RIPA lysate and protease inhibitor, and the tissue was ground. The samples were fully lysed at 4 °C for 30 min and centrifuged at 12,000 g for 15 min. The total protein concentration was determined by the BCA method. Protein denaturation was performed at 95 °C for 10 min. An amount of 30 μg protein was added to SDS-PAGE gel for electrophoresis. After separation, the protein was transferred to PVDF membranes and blocked with 5% skim milk for 1 h. The primary antibody was incubated at 4 °C overnight. The film was washed with TBST, and the second antibody was incubated for 1 h at room temperature. After the film was washed with TBST, ECL luminescent solution was added and photographed (Bio-Rad, Hercules, CA, USA). Image Lab software was used for statistical analysis. The antibodies used in this study are shown in Table 1.

### 4.10. Real-Time PCR

Total RNA was extracted from mouse liver tissue and cells by the TRIzol method and reversely transcribed into cDNA. Real-time PCR was performed in the CFX96 Touch™ instrument (CFX96 Touch™, Bio-Rad, Hercules, CA, USA). The reaction conditions were as follows: pre-denaturation was performed at 95 °C for 30 s, one cycle; denaturation at 95 °C, 5 s, 40 cycles; annealing extension at 60 °C, 30 s, 40 cycles. The relative expression levels of target genes were calculated by the 2^−ΔΔCT^ method according to the amplified CT values [52]. Primer sequences are shown in Table 2.

### 4.11. Immunofluorescent Staining

The tissue sections were deparaffinized and 3% H_2_O_2_ was used to block the endogenous peroxidase. The sections were incubated in tris-buffered saline (TBS) with 5% albumin bovine V (BSA; Solarbio, Beijing, China) for 1 h. The cell samples were washed with PBS three times and fixed with 4% paraformaldehyde for 15 min. After washing with PBS three times, tissue sections or cells were permeated with 0.3% Triton X-100 for 15 min and then incubated with p21 antibody (1:200; Abcam, Cambridge, MA, USA), p16 antibody (1:200; Cell Signaling Technology, Danvers, MA, USA), LC3 antibody (1:200; CST, Danvers, MA, USA), PCNA antibody (1:200; Proteintech, Wuhan, China) or Ki67 antibody (1:200; Abways, Shanghai, China) in 5% BSA overnight at 4 °C. After washing with TBS, the sections were incubated with a Rhodamine (TRITC)-conjugated goat anti-rabbit IgG (H+L) (1:300; Abcam, Cambridge, UK). The nuclei were counterstained with DAPI (Invitrogen, Waltham, MA, USA). The sections were washed with PBS three times, and coverslips were mounted in 90% glycerol in PBS. The fluorescence was detected by a fluorescence microscope (Nikon, Tokyo, Japan). Image J was used to analyze the fluorescence intensity.

### 4.12. Statistical Analyses

All data were presented as means ± standard deviation. Statistical analysis was performed using GraphPad Prism 7 (GraphPad Software, Inc, San Diego, CA, USA). Comparisons between two-groups were made with an unpaired *t*-test. Multiple group comparisons were made using a one-way analysis of variance. Tukey’s test was used as a post hoc test for pairwise comparisons. The data that were not normally distributed were analyzed using nonparametric statistical analysis. All experiments were independently repeated three times. *p* < 0.05 was considered statistically significant.

## Figures and Tables

**Figure 1 ijms-23-08267-f001:**
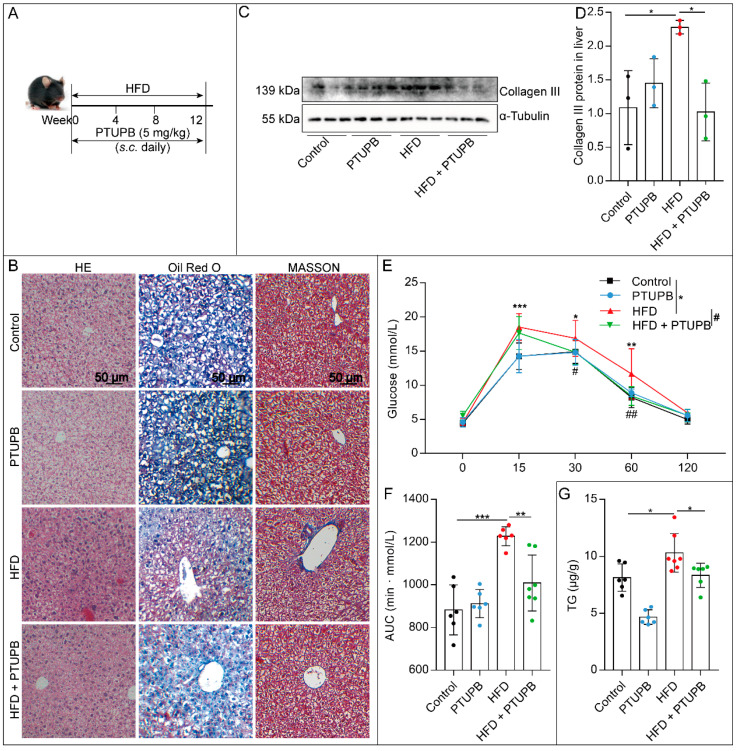
PTUPB reduces HFD-induced NAFLD in mice. PTUPB (5 mg/kg/day, *s.c.*) was administered daily (**A**). HE-stained histological images of liver sections from mice (**B**), left column, scale bars = 50 μm, *n* = 6. Oil Red O staining of liver sections from mice (**B**), middle column, scale bars = 50 μm, *n* = 6. Masson staining of liver sections from mice (**B**), right column, scale bars = 50 μm, *n* = 6. Western blot was used to detect collagen III protein expressions and quantitative analysis ((**C**, **D**), *n* = 6–7). Measurement of blood glucose and AUC during IGTT of mice fed with HFD or normal chow diet for 12 weeks ((**E**,**F**), *n* = 6–7). Content of TG in liver tissue of mice ((**G**), *n* = 6–7). Data are expressed as the mean ± SD. Differences among multiple groups were performed using ANOVA. Tukey’s test was used as a post hoc test for pairwise comparisons. # or * *p* < 0.05, ** *p* < 0.01, and *** *p* < 0.001.

**Figure 2 ijms-23-08267-f002:**
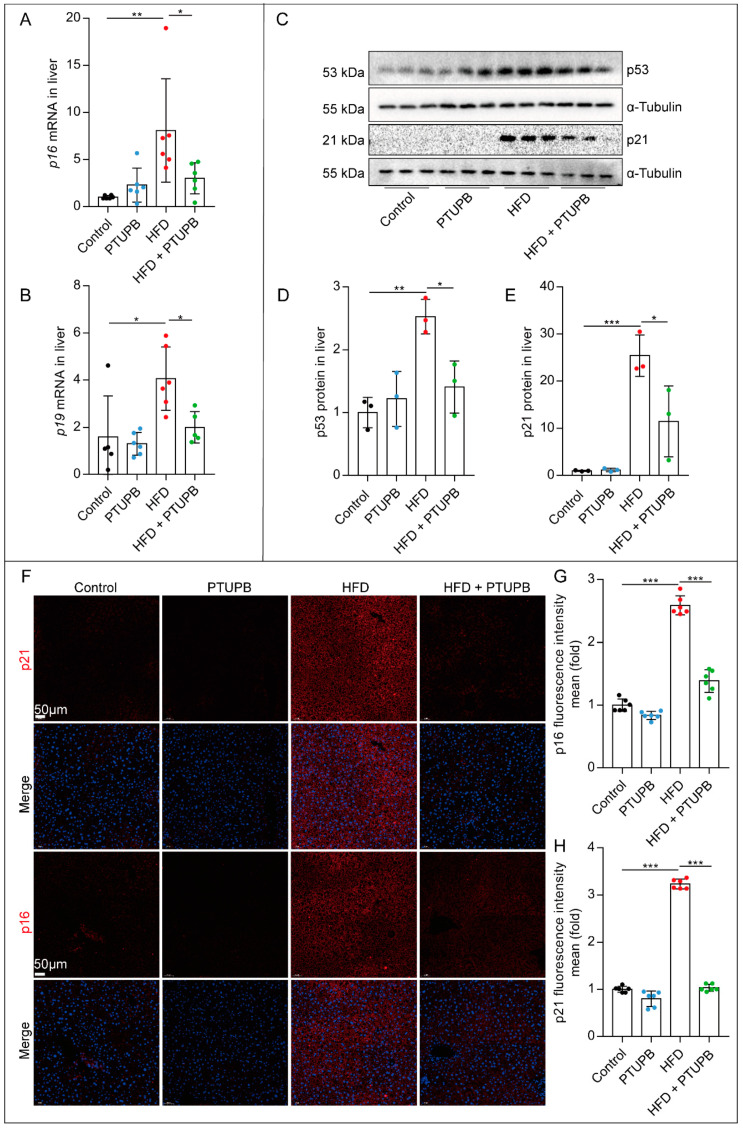
PTUPB inhibits hepatocyte senescence induced by HFD in mice. *p16* and *p19* mRNA expressions in the liver were detected using real-time PCR ((**A**,**B**), *n* = 5–6). Western blot was used to detect p53 and p21 protein expressions and quantitative analysis ((**C**–**E**), *n* = 6). The fluorescence intensity of p21 and p16 were detected by immunofluorescence ((**F**), scale bars = 50 μm, *n* = 6). Image J was used to analyze the fluorescence intensity of p21 and p16 ((**G**,**H**), *n* = 6). Data are expressed as the mean ± SD. Differences among multiple groups were performed using ANOVA. Tukey’s test was used as a post hoc test for pairwise comparisons. * *p* < 0.05, ** *p* < 0.01, and *** *p* < 0.001.

**Figure 3 ijms-23-08267-f003:**
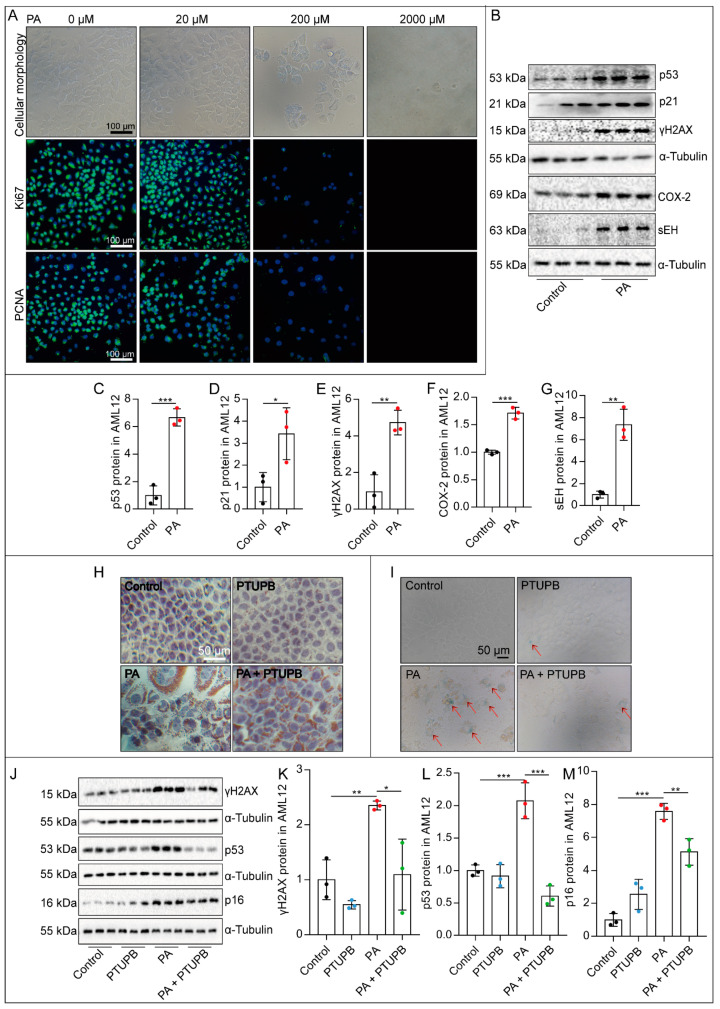
PTUPB attenuates PA-induced hepatocyte senescence *in vitro*. AML12 cells were treated with a series of concentrations of PA (0, 20, 200, and 2000 μM) for 48 h, and then the changes in cell morphology and number were photographed under a microscope ((**A**), scale bars = 100 μm, *n* = 3). The fluorescence intensity of Ki67 and PCNA was detected by immunofluorescence ((**A**), scale bars = 100 μm, *n* = 3). AML12 cells were treated with PTUPB (1 μM) for 1 h before treatment with PA (200 µM). Forty-eight hours after the PA administration, the protein expressions of p53, p21, γH2AX, COX-2, and sEH in AML12 cells were measured by Western blot and quantitatively analyzed ((**B**–**G**), *n* = 3). Representative images of Oil Red O staining in AML-12 hepatocytes treated with PA with/without PTUPB for 48 h ((**H**), bar = 50 µm). Senescence was confirmed by SA-β-gal staining ((**I**), bar = 50 µm). The protein expressions of γH2AX, p53, and p16 in AML12 cells were measured by Western blot and quantitatively analyzed after PA stimulation with/without PTUPB for 48 h ((**J**–**M**), *n* = 3). Data are expressed as the mean ± SD. Differences among multiple groups were performed using ANOVA. Tukey’s test was used as a post hoc test for pairwise comparisons. Comparisons between the two-group were made with an unpaired *t*-test. * *p* < 0.05, ** *p* < 0.01, and *** *p* < 0.001.

**Figure 4 ijms-23-08267-f004:**
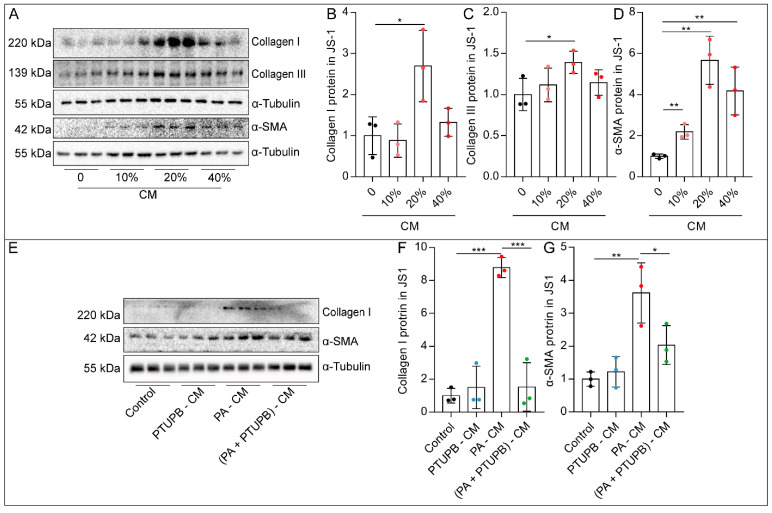
PTUPB reduces hepatic astrocyte activation by inhibiting hepatocyte senescence. AML12 cells were treated with PA (200 µM) for 48 h, and cell culture supernatant was collected as CM. The protein expressions of collagen I, collagen III, and α-SMA in JS1 cells treated with different concentrations of PA-CM stimulation for 48 h were measured by Western blot and quantitatively analyzed ((**A**–**D**), *n* = 3). The protein expressions of Collagen I and α-SMA in JS1 cells were measured by Western blot and quantitatively analyzed after different CM stimulation for 48 h ((**E**–**G**), *n* = 3). Data are expressed as the mean ± SD. Differences among multiple groups were performed using ANOVA. Tukey’s test was used as a post hoc test for pairwise comparisons. * *p* < 0.05, ** *p* < 0.01, and *** *p* < 0.001.

**Figure 5 ijms-23-08267-f005:**
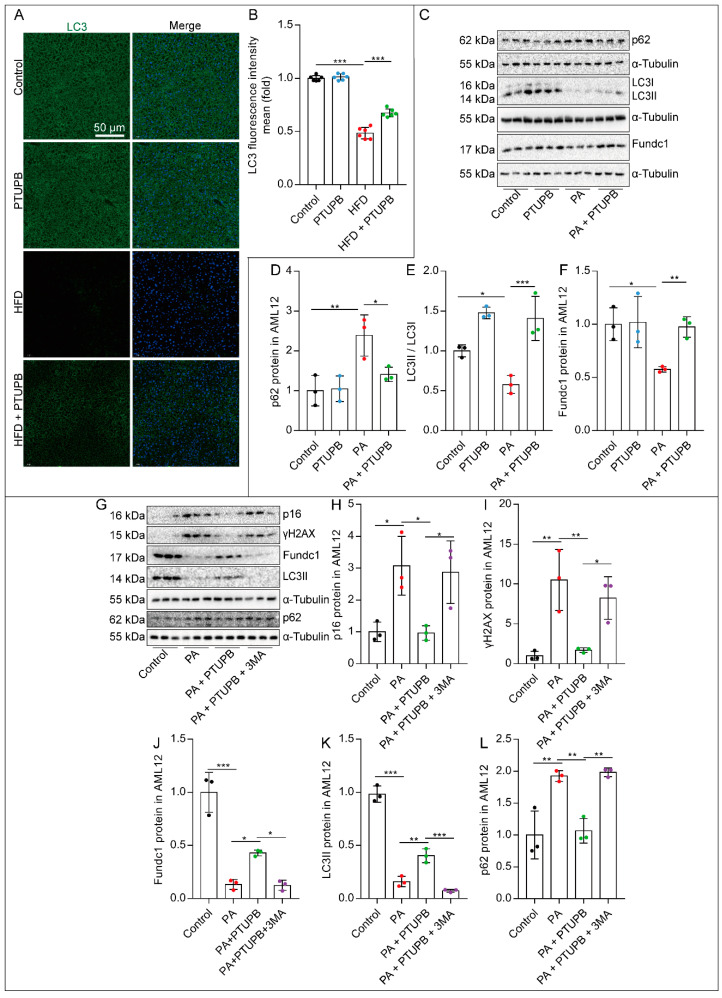
PTUPB attenuates hepatocyte senescence by enhancing autophagy in AML12 cells. The fluorescence intensity of liver LC3 was detected by immunofluorescence ((**A**), scale bars = 50 μm, *n* = 6). Image J was used to analyze the fluorescence intensity of LC3 ((**B**), *n* = 6). Cells were treated with PTUPB (1 μM) for 1 h before treatment with PA (200 µM). The protein expressions of p62, LC3I/II, and Fundc1 were measured by Western blot and quantitatively analyzed in AML12 cells treated with PA for 48 h ((**C**–**F**), *n* = 3). Cells were treated with PTUPB (1 μM) and 3-MA (5 mM) for 1 h before treatment with PA (200 µM). The protein expressions of p16, γH2AX, Fundc1, LC3II, and p62 were measured by Western blot and quantitatively analyzed in AML12 cells ((**G**–**L**), *n* = 3). Data are expressed as the mean ± SD. Differences among multiple groups were performed using ANOVA. Tukey’s test was used as a post hoc test for pairwise comparisons. * *p* < 0.05, ** *p* < 0.01, and *** *p* < 0.001.

**Figure 6 ijms-23-08267-f006:**
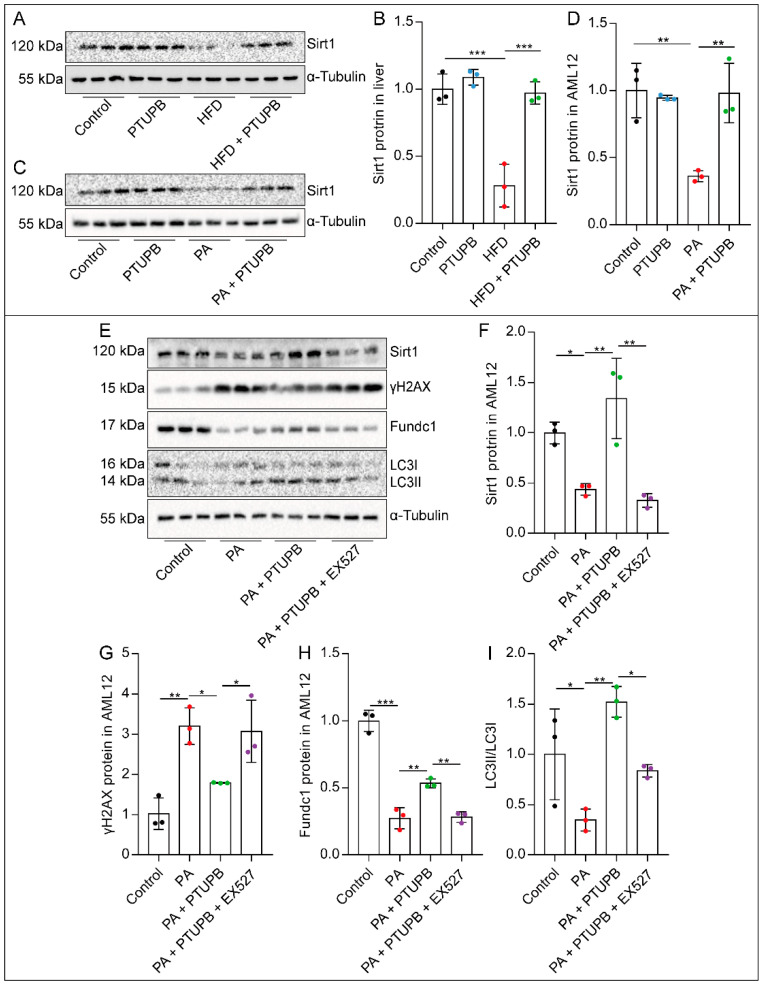
PTUPB enhances hepatocyte autophagy by promoting Sirt1 expression. The protein expressions of Sirt1 in the liver ((**A**,**B**), *n* = 6) and AML12 cells ((**C**,**D**), *n* = 3) were measured by Western blot and quantitatively analyzed. Cells were treated with PTUPB (1 μM) or EX527 (10 μM) for 1 h before treatment with PA (200 µM) for 48 h. The protein expressions of Sirt1, γH2AX, Fundc1, and LC3II/I in AML12 cells were measured by Western blot and quantitatively analyzed ((**E**–**I**), *n* = 3). Data are expressed as the mean ± SD. Differences among multiple groups were performed using ANOVA. Tukey’s test was used as a post hoc test for pairwise comparisons. * *p* < 0.05, ** *p* < 0.01, and *** *p* < 0.001.

**Figure 7 ijms-23-08267-f007:**
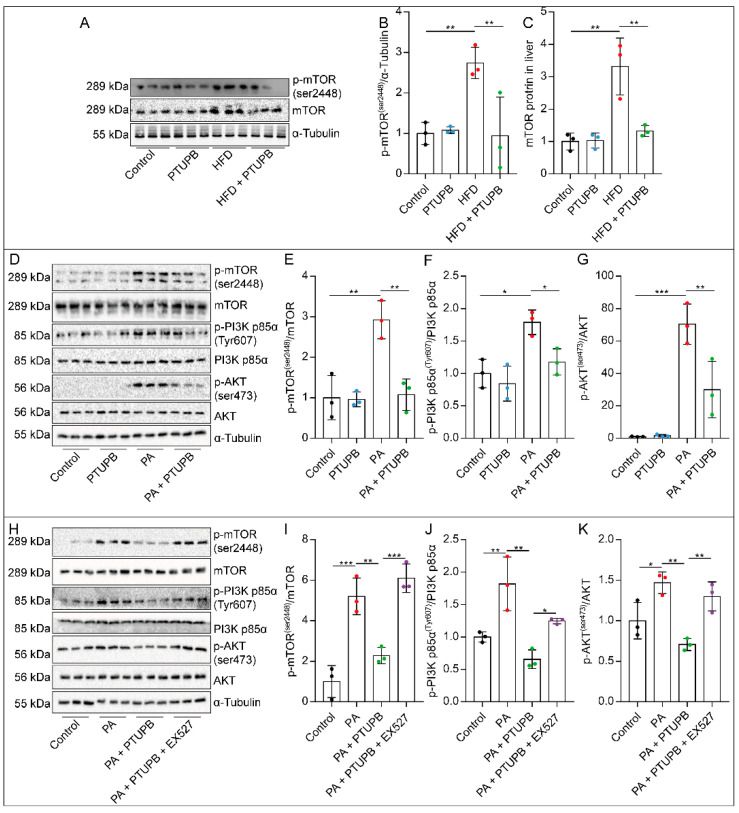
PTUPB inhibits the PI3K/AKT/mTOR signaling pathway by promoting Sirt1 expression. The mTOR protein and its phosphorylation level in the liver were measured by Western blot and quantitatively analyzed ((**A**–**C**), *n* = 6). Cells were treated with PTUPB (1 μM) for 1 h before treatment with PA (200 µM). The levels of p-mTOR, p-AKT, and p-PI3K in AML12 cells were measured by Western blot and quantitatively analyzed after PA stimulation for 30 min ((**D**–**G**), *n* = 3). Cells were treated with PTUPB (1 μM) or EX527 (10 μM) for 1 h before treatment with PA (200 µM). The levels of p-mTOR, p-AKT, and p-PI3K in AML12 cells were measured by Western blot and quantitatively analyzed after PA stimulation for 30 min ((**H**–**K**), *n* = 3). Data are expressed as the mean ± SD. Differences among multiple groups were performed using ANOVA. Tukey’s test was used as a post hoc test for pairwise comparisons. * *p* < 0.05, ** *p* < 0.01, and *** *p* < 0.001.

**Figure 8 ijms-23-08267-f008:**
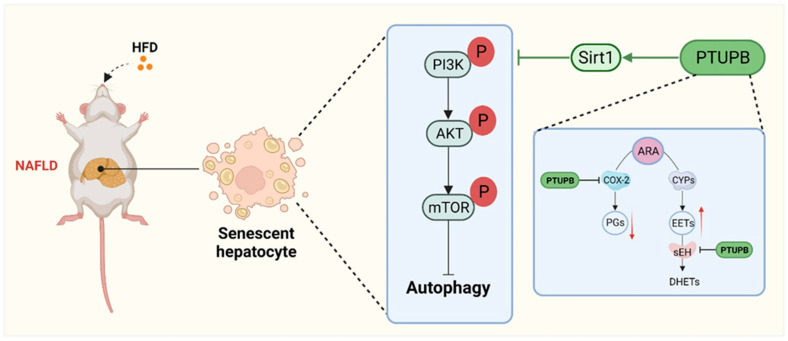
Schematic illustration. PTUPB alleviates HFD-induced NAFLD by reducing hepatocyte senescence through autophagy in mice.

**Table 1 ijms-23-08267-t001:** Antibodies were used in this study.

Antibodies	Source	Catalog
Anti-collagen III polyclonal antibody	Proteintech	22734-1-AP
Anti- collagen I monoclonal antibody	CST	#84336
Anti- αSMA polyclonal antibody	SAB	41550
Anti-p53 polyclonal antibody	Proteintech	10442-1-AP
Anti-p21 polyclonal antibody	Servicebio	GB11153
Anti-p16 monoclonal antibody	Abcam	ab211542
Anti-COX-2 polyclonal antibody	Proteintech	12375-1-AP
Anti-sEH monoclonal antibody	Abcam	ab155280
Anti-γH2AX monoclonal antibody	Boster	BM4841
Anti-LC3 monoclonal antibody	CST	12741
Anti-Fundc1 monoclonal antibody	CST	49240
Anti-p62 polyclonal antibody	Immunoway	YT7058
Anti-ki67 polyclonal antibody	Abways	CY5542
Anti-PCNA polyclonal antibody	Proteintech	10205-2-AP
Anti-Sirt1 monoclonal antibody	Proteintech	60303-1-Ig
Anti-mTOR (phospho ser2448) antibody	CST	2972S
Anti-mTOR polyclonal antibody	Proteintech	20657-1-AP
Anti-PI3K p85α (phospho Tyr607) antibody	Affinity	AF3241
Anti-PI3K p85α monoclonal antibody	Immunoway	YM3503
Anti- AKT (phospho ser473) polyclonal antibody	Immunoway	YP0846
Anti-AKT polyclonal antibody	Immunoway	YT0185
Anti-α-Tubulin polyclonal antibody	Servicebio	GB11200

**Table 2 ijms-23-08267-t002:** Sequences of specific primers were used in this study.

Gene	Forward Primer (5′-3′)	Reverse Primer (5′-3′)
*m-p16*	CTCTGCTCTTGGGATTGGC	GTGCGATATTTGCGTTCCG
*m-p19*	GAGGCCGGCAAATGATCATAGA	GTGGATACCGGTGGACTGTG
*m-β-actin*	TTCCAGCCTTCCTTCTTG	GGAGCCAGAGCAGTAATC

## Data Availability

The datasets generated during and/or analyzed during the current study are available from the corresponding author upon reasonable request.

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
