# Peer review of "COX-2/sEH Dual Inhibitor Alleviates Hepatocyte Senescence in NAFLD Mice by Restoring Autophagy through Sirt1/PI3K/AKT/mTOR"

_ijms, 2022, doi:10.3390/ijms23158267_

Round 1

Reviewer 1 Report

The manuscript is an interesting original research investigating whether a COX-2/she dual inhibitor treatment is able to restore the hepatocytes damage in a HDF-induced NAFLD mice and in vitro models. This work contributes to the field of study reporting an PTUPB-mediated anti-senescence effect downregulating p53, p21, p16 and gH2AX protein expression, along with an inhibition of hepatic astrocytes activation and induction of autophagy that could be promoted by Sirt1 upregulation. However, the evidence for the autophagy stimulation is weak and the analysis of p62, at least, should be included. In addition, the manuscript needs some improvements for publication.

Major changes:

1.     The authors should include the analysis of, at least, p62 autophagy marker and Beclin-1 or p-ULK1 or some ATG proteins, so they can get stronger evidence to state that the PTUPB treatment restores autophagy in their NAFLD model. As indicates the guides to monitoring autophagy flux, Western blot LC3-II is not enough to affirm a change in autophagic flux, so if different tests are not included to monitor it, such as flow cytometry or confocal images, it should at least include Western blots for autophagy-specific substrates such as p62.

2.     The authors should include the analysis of Sir1 and mTOR pathway in the mice model.

3.     Describe any criteria used for including or excluding mice during the experiment, and data points during the analysis. Specify if these criteria were established a priori. If no criteria were set, state this explicitly.

4.     For each experimental group, report any mice or data points not included in the analysis and explain why. If there were no exclusions, state so.

5.     Was the data check for normal distribution? If so, the authors should include the information about the test in the statistical analysis section.

Minor changes

1.     The authors should indicate the information about the microscope and the camera used (2.6 and 2.7).

2.     The authors should correct some misspellings like “FDH” (abstract), missing spaces (8 to10; 37ºC, 95ºC; EX527(Fig.) and tables distribution.

Reviewer 2 Report

Comments on the article entitled: “COX-2/sEH dual inhibitor alleviates hepatocyte senescence in NAFLD mice by restoring autophagy through Sirt1/PI3K/AKT/mTOR” By Chen-Yu Zhang and collaborators.

The Authors investigated whether treatment with PTUPB (a dual inhibitor of sEH and COX-2) could reduce obesity-induced hepatocyte senescence, and studied at the effect of PTUPB on senescence and autophagy

Specific comments:

Table 1: the column Source should be reformatted to allow alignment with the antibodies

3. Results Section:

Section 3.1: Figure 1B: A higher magnification of the images are needed to show the signals. 

What is the graph next to panel E. Please clarify and describe this result.

Section 3.2: Figure 2B, the signal for p21 on the western is not a sharp and we cannot detect a band but rather a smear. Therefore, the quantification is questionable. The blots should be repeated.

Figure 2B, I cannot see any signal. Please replace or improve the quality of all images.

The graphs for the quantification of p53 and p21 representative of the westerns should be labeled D and E and indicated in the legend.

Section 3.3: The quality of the images of Figure 3A are so bad I cannot see anything. Please replace higher magnification and better resolution. Consequently, I cannot comment on the cell morphological changes. Anyway, to prove that the cells are growth arrest Ki67 staining is required.

The graphs regarding the protein quantification should be labeled and indicated in the figure legend.

Again, Figure 3D higher magnification is need to detect the signals and cellularity.

Figure 3E image are blurry cannot see anything

Figure 3F, the western blot signal for p53 is poor and difficult to judge how the quantification has been performed.

Section 3.4Figure 4A. The western detection of Col1 and Col3 are so bad, I cannot understand how the quantification can be done accurately. Consequently, the graphs not labeled next to panel A are probably not correct.

Same comment for the Western Figure 4B regarding collagen 1

Section 3.5: Figure 5A: I cannot see any of the images. Please improve the quality and resolution.

An autophagy activity assay should be performed to evaluate the extend to which this activity is reduced.

Overall, the quality of the figures and their results need to be highly improved to support the conclusions reported in this manuscript.

Reviewer 3 Report

The authors studied the mechanism of action of soluble epoxide hydrolase (sEH)/cyclooxygenase-2 (COX-2)-mediated arachidonic acid (ARA) metabolism that contributes to the pathogenesis of non-alcoholic fatty liver disease (NAFLD). in mice. The authors point to the essential role of cellular senescence in NAFLD based on accumulating evidence. In this manuscript, the authors investigated whether restoring the balance of sEH/COX-2-mediated ARA metabolism attenuated NAFLD through hepatocyte senescence. To do this, they have used a dual inhibitor of sEH and COX-2, PTUPB, to restore the balance of ARA mediated by sEH/COX-2 metabolism. They have performed in vivo experiments, in which they induced NAFLD with a high-fat diet (HFD) using C57BL/6J mice. invite, the authors used mouse hepatocytes (AML12) and mouse hepatic astrocytes (JS1) to investigate the effects of PTUPB on hepatocyte senescence induced by palmitic acid (PA) and its mechanism. The authors observed that PTUPB relieved liver injury, decreased collagen and lipid accumulation, restored glucose tolerance, and decreased Hepatic triglyceride levels in HFD-induced NAFLD mice. In addition, PTUPB significantly reduced the expression of p16, p53, and p21 molecules that are markers of senescence in the liver of FDH mice. In vitro experiments revealed that PTUPB decreased the lipid content, the number of β-gal positive cells, and the expression of p53, p21, and γH2AX proteins in PA-treated AML12 cells. For its part, PTUPB reduced the activation of JS1 hepatic astrocytes by slowing down the senescence of AML12 cells in a coculture system. It was further observed that PTUPB enhances the autophagy-related protein LC3II/I ratio in AML12 cells by upregulating Fundc1 protein expression and suppressing hepatocyte senescence. In addition, PTUPB enhanced hepatocyte autophagy by inhibiting the PI3K/AKT/mTOR pathway through Sirt1, which contributed to the suppression of senescence. The authors conclude that PTUPB inhibits PI3K/AKT/mTOR via Sirt1, enhancing autophagy and slowing hepatocyte senescence, alleviating NAFLD.

 Minor Concerns:

1-The authors show in Figure 2C immunofluorescence images where neither p21 nor p16 intensity is observed. Could the authors attach some images where what the authors indicate in the graph attached to the figure can be observed?

2-The anti-Collagen III incubated membrane of Figure 4A is blurred. The authors should show a membrane where the antibody signal is clearly visible.

Round 2

Reviewer 1 Report

Comments and suggestions have been taken into account and errors have been modified, so the manuscript has improved considerably and is ready for publication.